# Transcendence: Generative Models Can Outperform The Experts That Train Them

**Edwin Zhang**
OpenAI
Harvard University
Humanity Unleashed
edwin@openai.com

**Vincent Zhu**
UC Santa Barbara
Humanity Unleashed
vincentzhu@ucsb.edu

**Naomi Saphra**
Harvard University
Kempner Institute
nsaphra@g.harvard.edu

**Anat Kleiman**
Harvard University
Apple
anatkleiman@g.harvard.edu

**Benjamin L. Edelman**
Princeton University
Harvard University
bedelman@g.harvard.edu

**Milind Tambe**
Harvard University
tambe@g.harvard.edu

**Sham Kakade**
Harvard University
Kempner Institute
sham@g.harvard.edu

**Eran Malach**
Harvard University
Kempner Institute
emalach@g.harvard.edu

## Abstract

Generative models are trained with the simple objective of imitating the conditional probability distribution induced by the data they are trained on. Therefore, when trained on data generated by humans, we may not expect the artificial model to outperform the humans on their original objectives. In this work, we study the phenomenon of *transcendence*: when a generative model achieves capabilities that surpass the abilities of the experts generating its data. We demonstrate transcendence by training an autoregressive transformer to play chess from game transcripts, and show that the trained model can sometimes achieve better performance than all players in the dataset.[1] We theoretically prove that transcendence can be enabled by low-temperature sampling, and rigorously assess this claim experimentally. Finally, we discuss other sources of transcendence, laying the groundwork for future investigation of this phenomenon in a broader setting.

## 1   Introduction

Generative models (GMs) are typically trained to mimic human behavior. These humans may be skilled in their various human objectives: answering a question, creating art, singing a song. The model has only one objective: minimizing the cross-entropy loss with respect to the output distribution, thereby adjusting it to match the distribution of human labels[2]. Therefore, one might assume the model can, at best, match the performance of an expert on their human objectives. Is it possible for these models to surpass—to *transcend*—their expert sources in some domains?

---

[1]To play with our models, code, and data, please see our website at https://transcendence.eddie.win.

[2]Although chatbots are subject to a variety of post-training tuning methods, e.g., RLHF, we restrict our scope by assuming that the specialized knowledge and capacities are already provided by cross-entropy loss.

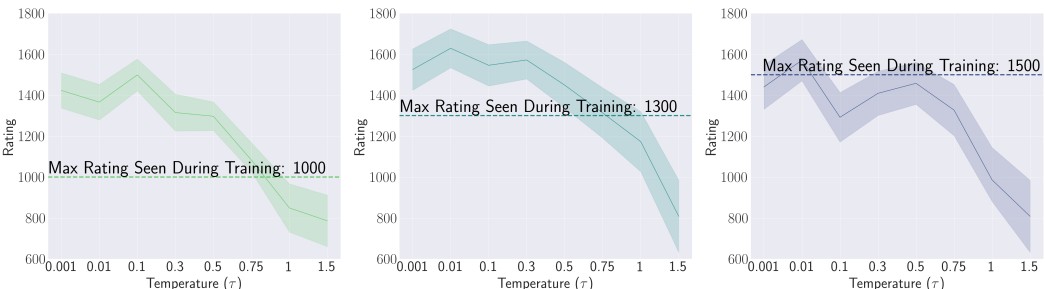

Figure 1: Ratings of our autoregressive decoder-only transformer, ChessFormer, over several different temperatures. We refer to our models as "ChessFormer <Maximum Glicko-2 rating seen during training>" to easily distinguish between different models in subsequent sections. Each model is trained only on games with players up to a certain rating (1000, 1300, 1500, respectively). We report 95% confidence intervals calculated through taking $\pm 1.96\sigma$.

We illustrate an example of such *transcendence* in Figure 1, which measures the chess ratings (Glicko-2 [7]) of several transformer [35] models. Our experimental testbed is generative modeling on chess, which we choose as a domain for its well-understood, constrained nature. The transformer models are trained on public datasets of human chess transcripts, autoregressively predicting the next move in the game. To test for *transcendence*, we limit the maximal rating of the human players in the dataset below a specified score. We find that ChessFormer 1000 and ChessFormer 1300 (the latter number being the maximum rating seen during training) achieve significant levels of transcendence, surpassing the maximal rating seen in the dataset. Our focus is this capacity of a GM to transcend its expert sources by broadly outperforming any one expert. The key to our findings is the observation that GMs implicitly perform *majority voting* over the human experts. As these models are trained on a collection of many experts with diverse capacities, predilections, and biases, this majority vote oftentimes outperforms any individual expert, a phenomena that is known as "wisdom of the crowd".

Our objective is to formalize the notion of transcendence and focus narrowly on this source of improvement over the experts: the removal of diverse human biases and errors. We prove that this form of denoising is enabled by low-temperature sampling, which implicitly induces a majority vote. Our result draws a subtle but deep connection from our new setting to a rich prior literature on model ensembling [1, 6, 19], enabling several key results. We precisely characterize the conditions under which *transcendence* is possible, and give a rigorous theoretical framework for enabling future study into the phenomenon. To test the predictive power of our theory, we then empirically demonstrate these effects. Digging deeper into the effects of majority voting, we show that its advantage is primarily due to performing much better on a small subset of states—that is, under conditions that are likely key to determining the outcome of the game. We also find that diversity in the data is a necessary condition for practically effective majority voting, confirming our theoretical findings. In short:

- We formalize the notion of transcendence in generative models (Section 2).

- We find a key insight explaining one cause of transcendence by connecting the case of denoising experts to model ensembling. In low temperature sampling settings, we prove that a generative model can transcend if trained on a single expert that makes mistakes uniformly at random. We then extend this result to transcending a collection of experts that are each skilled in different domains (Section 3).

- We train a chess transformer on game transcripts that only include players up to a particular skill level. We confirm our theoretical prediction that this model only surpasses the maximum rating of its expert data generators at low temperature settings (Section 4).

- We visualize the distribution of changes in reward by setting a lower sampling temperature, attributing the increased performance to large improvements on a relatively small portion of states (Section 4.2).

- We explore the necessity of dataset diversity, and the inability of ChessFormer to transcend when trained on less diverse datasets (Section 4.2).

## 2 Definition of Transcendence

Denote by $\mathcal{X}$ the (variable-length) input space and by $\mathcal{Y}$ the (finite) output space. Let $\mathcal{F}$ be the class of all functions mapping $\mathcal{X} \mapsto P(\mathcal{Y})$ (where we use the notation $P(\mathcal{Y})$ to denote probability distributions over $\mathcal{Y}$). That is, the functions in $\mathcal{F}$ map inputs in $\mathcal{X}$ to probability distributions over $\mathcal{Y}$, so each function $f \in \mathcal{F}$ defines a conditional probability distribution of $y \in \mathcal{Y}$ given $x \in \mathcal{X}$. We denote this distribution by $f(y|x)$.

Fix some input distribution $p$ over $\mathcal{X}$ such that $p$ has full support (namely, for every $x \in \mathcal{X}$ we have $p(x) > 0$). Throughout the paper, we assume that our data is labeled by $k$ experts, denoted $f_1,...,f_k \in \mathcal{F}$. Namely, we assume that the inputs are sampled from the input distribution $p$ and then each input $x \in \mathcal{X}$ is labeled by some expert chosen uniformly at random[3]. This process induces a joint probability distribution over $\mathcal{X} \times \mathcal{Y}$, which we denote by D. Specifically, $\mathrm{D}(x,y) = p(x)\overline{f}(y|x)$ where $\overline{f}$ is the mixture of the expert distributions, namely

$$\overline{f}(y|x) = \frac{1}{k}\sum_{i=1}^{k} f_i(y|x) \tag{1}$$

We measure the quality of some prediction function $f \in \mathcal{F}$ using a reward assigned to each input-output pair. Namely, we define a reward function $r : \mathcal{X} \times \mathcal{Y} \to \mathbb{R}$, s.t. for all $x$, the function $r(x,\cdot)$ is not constant (i.e., for every input $x$ not all outputs have the same reward). We choose some test distribution $p_{\text{test}}$ over $\mathcal{X}$, and for some $f \in \mathcal{F}$ define the average reward of $f$ over $p_{\text{test}}$ by:

$$R_{p_{\text{test}}}(f) = \mathbb{E}_{x \sim p_{\text{test}}}[r_x(f)], \quad \text{where } r_x(f) = \mathbb{E}_{y \sim f(\cdot|x)}[r(x,y)] \tag{2}$$

A learner has access to the distribution D, and needs to find a function that minimizes the cross-entropy loss over D. Namely, the learner chooses some function $\hat{f} \in \mathcal{F}$ s.t. $\hat{f} = \mathrm{argmin}_{f \in \mathcal{F}}\mathbb{E}_{x \sim p}\left[H(\overline{f},f)\right]$ where $H$ is the cross-entropy function.

**Definition 1.** *We define "transcendence" to be a setting of $f_1,...,f_k \in \mathcal{F}$ and $p \in P(\mathcal{X})$ where:*

$$R_{p_{\text{test}}}(\hat{f}) > \max_{i \in [k]} R_{p_{\text{test}}}(f_i) \tag{3}$$

In other words, transcendence describes cases where the learned predictor performs better (achieves better reward) than the best expert generating the data. Note that we are focusing on an idealized setting, where the learner has access to infinite amount of data from the distribution D, and can arbitrarily choose any function to fit the distribution (not limited to a particular choice of architecture or optimization constraints). As we will show, even in this idealized setting, transcendence can be impossible to achieve without further modifying the distribution.

**Remark 1.** *We have made various simplifying assumptions when introducing our setting. For example, we assume that all experts share the same input distribution, we assume that all inputs have non-zero probability under the training distribution p, and we assume the experts are sampled uniformly at random. We leave a complete analysis of a more general setting to future work, and discuss this point further in section 6.*

## 3 Conditions for Transcendence

In this section we analyze the necessary and sufficient conditions for transcendence in our setting. We begin by showing that low-temperature sampling is *necessary* for transcendence in our specific setting. Then, we analyze specific sufficient conditions for transcendence, both in the case where the data is generated by a single expert and when the data is generated by multiple experts. We defer all proofs to Appendix A.

### 3.1 Low-Temperature Sampling is Necessary for Transcendence

Observe that by definition of $\hat{f}$, and using standard properties of the cross-entropy loss, we get that $\hat{f} = \overline{f}$, as defined in Eq. (1). Therefore, the conditional probability distribution generated by $\hat{f}$ is simply an average of the distributions generated by the expert. Since the reward is a linear function of these distributions, we get that $\hat{f}$ never achieves transcendence:

---

[3]Equivalently, we can assume that each example is labeled by all experts.

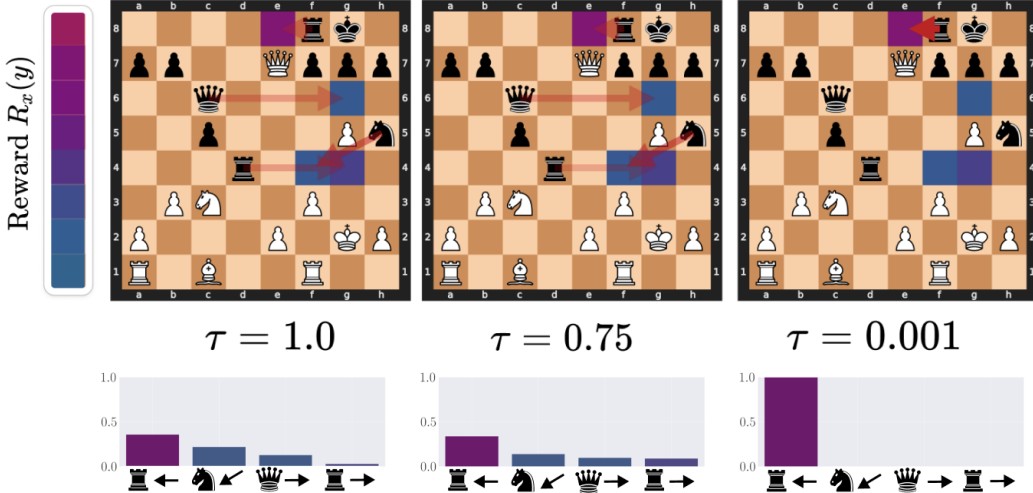

Figure 2: Visualizing the denoising effects of low temperature on the action distribution: an example of Chess-Former shifting probability mass towards the high reward move of trapping the queen with the rook as the temperature $\tau$ decreases. Opacity of the red arrows represent the probability mass given to different moves. The color of the square represent the reward that would be given for taking the action that moves the given piece to that state. Purple here is high reward, while blue is low. For more visualizations, see Appendix B.

**Proposition 1.** *For all choice of $f_1,...,f_k$ and $p_{\text{test}}$, there exists some $f_i$ s.t. $R_{p_{\text{test}}}(f_i) \geq R_{p_{\text{test}}}(\hat{f})$.*

Note that in our setting, we assume that all experts are sampled uniformly for a given input $x$. If instead this assumption is removed, then it may be possible to achieve transcendence with a bayesian weighting. We leave this analysis for future work.

### 3.2 Transcendence with Low-Temperature Sampling

Now, we consider a temperature sampling scheme over the learned function $\hat{f}$. Namely, for some temperature $\tau > 0$, and some probability distribution $q \in P(\mathcal{Y})$, denote the softmax operator with temperature $\tau$ by $\text{softmax}(q; \tau) \in P(\mathcal{Y})$ s.t. $\text{softmax}(q; \tau)_y = \dfrac{\exp(q_y/\tau)}{\sum_{y' \in \mathcal{Y}} \exp(q_{y'}/\tau)}$. Additionally, we define $\text{argmax}(q) \in P(\mathcal{Y})$ to be the uniform distribution over the maximal values of $q$, namely $\text{argmax}(q) = 1/|Y_q|$ if $y \in Y_q$ and $0$ if $y \notin Y_q$, where $Y_q = \{y \in \mathcal{Y} : q_y = \max(q)\}$. Now, define $\hat{f}_\tau$ to be the temperature sampling of $\hat{f}$, i.e. $\hat{f}_\tau(\cdot|x) = \text{softmax}(\hat{f}(\cdot|x); \tau)$ and $\hat{f}_{\max}$ the arg-max "sampling" of $\hat{f}$, i.e. $\hat{f}_{\max}(\cdot|x) = \text{argmax}(\hat{f}(\cdot|x))$. We now show that if the arg-max predictor $\hat{f}_{\max}$ is better than the best expert, then transcendence is possible with low-temperature sampling.

**Proposition 2.** $R_{p_{\text{test}}}(\hat{f}_{\max}) > \max_{i \in [k]} R_{p_{\text{test}}}(f_i)$ *if and only if there exists some temperature* $\tau \in (0,1)$ *s.t. for all* $0 \leq \tau' \leq \tau$, *it holds that* $R_{p_{\text{test}}}(\hat{f}_{\tau'}) > \max_{i \in [k]} R_{p_{\text{test}}}(f_i)$.

The above shows that, even though transcendence cannot be achieved when directly modeling the distribution, it can be achieved by temperature sampling, assuming that the arg-max predictor achieves higher reward compared to all experts. In other words, we make the subtle connection here that low-temperature sampling can be thought of as performning *majority vote* [1, 6] between the experts. Please see Appendix A for a formal proof of this connection. When the experts put non-negligible mass onto the best actions, the resulting majority vote may find the best action [9], which improves performance compared to individual experts (i.e., "wisdom of the crowd") and thus achieve transcendence.

### 3.3 Denoising a Single Expert

We now turn to study particular cases where low-temperature sampling can lead to transcendence. The most simple case is of a single expert that outputs a correct but noisy prediction. Denote by $f^*$ the optimal expert, s.t. for all $x$ we have $f^*(y|x) = \dfrac{\delta(y \in Y_x^*)}{|Y_x^*|}$, where $Y_x^* = \{y \in \mathcal{Y} : y = \max_{y'} r(x, y')\}$

and $\delta(\text{condition})$ is 1 if the condition is true and 0 otherwise. Now, for some $\rho \in (0,1)$, let $f_\rho$ be a "noisy" expert, s.t., for all $x$, with probability $\rho$ chooses a random output, and with probability $1-\rho$ chooses an output according to the optimal expert $f^*(\cdot|x)$, namely $f_\rho(y|x) = \rho/|\mathcal{Y}| + (1-\rho)f^*(y|x)$. We show that transcendence is achieved with low-temperature sampling for data generated by $f_\rho$:

**Proposition 3.** *Assume the data is generated by a single expert $f_\rho$. Then, there exists some temperature $\tau \in (0,1)$ s.t. for all $\tau' \leq \tau$, the predictor $\hat{f}_{\tau'}$ achieves "transcendence".*

### 3.4 Transcendence from Multiple Experts

Next, we consider the case where the dataset is generated by multiple experts that complement each other in terms of their ability to correctly predict the best output. For example, consider the case where the input space is partitioned into $k$ disjoint subsets, $\mathcal{X} = \mathcal{X}_1 \dot\cup ... \dot\cup \mathcal{X}_k$, s.t. the $i$-th expert performs well on the subset $\mathcal{X}_i$, but behaves randomly on other subsets. Namely, assume the expert $f_i$ behaves as follows: $f_i(y|x) = \left( \frac{\delta(y \in Y_x^*)\delta(x \in \mathcal{X}_i)}{|Y_x^*|} + \frac{\delta(x \notin \mathcal{X}_i)}{|\mathcal{Y}|} \right)$ where $Y_x^*$ is as previously defined and $\delta(\text{condition})$ is 1 if the condition is true and 0 otherwise. We show that, assuming that the test distribution $p_{\text{test}}$ is not concentrated on a single subset $\mathcal{X}_i$, we achieve transcendence with low-temperature sampling:

**Proposition 4.** *Let $p_{\text{test}}$ be some distribution s.t. there are at least two subsets $\mathcal{X}_i \neq \mathcal{X}_j$ s.t. $p_{\text{test}}(\mathcal{X}_i), p_{\text{test}}(\mathcal{X}_j) > 0$. Then, if the data is generated by $f_1, ..., f_k$, there exists some temperature $\tau \in (0,1)$ s.t. for all $\tau' \leq \tau$, the predictor $\hat{f}_{\tau'}$ achieves "transcendence".*

In order to build intuition for Proposition 4, see Appendix C for an intuitive diagram.

## 4 Experiments

To evaluate the predictive power of our impossibility result of transcendence with no temperature sampling (Proposition 1) as well as our result of transcendence from multiple experts with low temperature sampling (Proposition 2), we turn to modeling and training chess players. Chess stands out as an attractive option for several reasons. Chess is a well-understood domain and more constrained than other settings such as natural language generation, lending to easier and stronger analysis. Evaluation of skill in chess is also natural and well-studied, with several rigorous statistical rating systems available. In this paper, we use the Glicko-2 rating system [7], which is also adopted by https://lichess.org, the free and open-source online chess server from which we source our dataset.

### 4.1 Experimental Setup

**Training Details.** We trained several 50M parameter autoregressive transformer decoders following best practices from modern large model training, including a cosine learning rate schedule and similar batch size-learning rate ratios as prescribed by the OPT-175B team [37]. Our dataset consists of human chess games from the lichess.org open source database from January 2023 to October 2023. In total, this dataset contains approximately one billion games. In this setting, an expert is a specific individual player. To test for transcendence, we truncate this dataset by a maximum rating, so that during training a model only sees data up to a given rating. We train our model on the next-token prediction objective, and represent our chess games as Portable Game Notation (PGN) strings, such as `1.e4 e5 2.Nf3 Nc6 3.Bb5... 1/2-1/2`. Note that we do not give any rating or reward information during training—the only input the model sees are the moves and the outcome of the game. We tokenize our dataset at the 32-symbol character level. (For further details, see Appendix E.) Our model plays chess "blind"— without direct access to the board state—and, furthermore, is never explicitly given the rules of the game: at no point is play constrained to valid outputs for a given piece or board state. Nontrivial chess skill is therefore not straightforward to acquire, and if not for the surprising capabilities of modern large transformers, one might imagine such a model would fail to learn even the basic rules of playing chess. This blindfolded setting has also been studied by prior work [23, 30], as discussed further in section 5.

One gap between our theory and practice is that in our theory, we assume that each expert is defined over the entire input space $\mathcal{X}$. However, in the chess setting such full coverage is extremely unlikely to be the case after around move 15, as there are more unique chess games than atoms in the universe due to the high branching factor of the game tree. To address this gap, we visualize the latent representation of our model in Figure 3, where we find the model is able to capture meaningful semantics regarding both

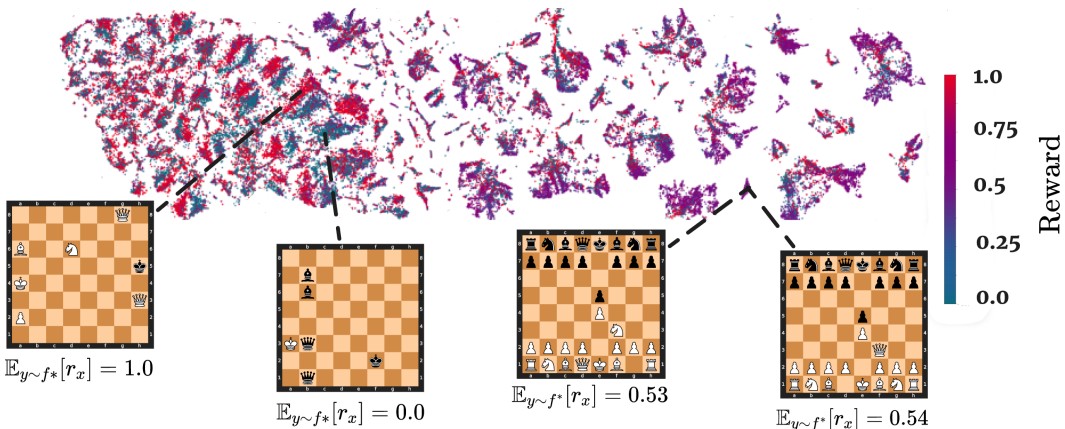

Figure 3: Inspired by Mnih et al. [20], we generate a t-SNE embedding [34] of ChessFormer's last hidden layer latent representations of game transcripts during training time. The colors represent the probability of winning, with +1 corresponding to a state where White has won and 0 to Black. Probabiliy of winning is computed through the Stockfish analysis engine. We also visualize several board states associated with different clusters in the t-SNE embedding, and their associated expected reward when following the expert Stockfish distribution. Note that the model distinguishes between states where the outcome has already been determined (the two left boards), versus opening states that are extremely similar (the two right boards). See the full t-SNE in Appendix G.

the relative advantage of a state, as well as the identity of the black and white player. This visualization illustrates the ability of our model to generalize by compressing games into some shared latent representation, enabling experts to generalize to unseen states, bridging this gap between theory and practice.

**Evaluation.**    We evaluate each model by its Glicko-2 ratings against Stockfish 16.1 [29], a popular open-source chess engine. Stockfish uses a traditional minimax search equipped with a bespoke CPU-efficient neural network for evaluation [22] and $\alpha$-$\beta$ pruning for further efficiency. We evaluate Stockfish at levels 1, 3, and 5 with a 100ms timeout directly on Lichess' platform against the Maia [18] 1, 5, and 9 bots (human behavior cloned convolutional networks trained at rating bins 1100-1200, 1500-1600, and 1900-2000, respectively) for several hundred games, obtaining calibrated Glicko-2 ratings for Stockfish specifically on Lichess' platform ($1552 \pm 45.2$, $1842 \pm 45.2$, $2142 \pm 59$ for Stockfish Levels 1, 3, and 5, respectively). Next, for evaluating our own models, we then play against Stockfish levels of 1, 3, and 5 for 100 games each, reaching a final rating calculation with 300 games. We then report both the Glicko-2 rating $R$ as well as rating deviation $RD$ of our models, where $R \pm 2 * RD$ provides a $95\%$ confidence interval. To play against Stockfish, we successively prompt our model with the current game PGN string. Note that our output is entirely unconstrained, and may be either illegal in the current board state or altogether unparsable. If our model fails to generate a valid legal move after 5 samples, we consider it to have lost. After generation, we give the updated board state to Stockfish and pass a new PGN string appended with the prior move of Stockfish back to our model. We repeat this process until the game ends.

## 4.2    Experimental Results

**Main Result: Low-temperature sampling enables transcendence.**    In this section we attempt to answer our primary research question, can low-temperature sampling actually induce *transcendence* in practice? We test Proposition 2 by evaluating several ChessFormers across different temperature values, from $0.001$ (nearly deterministic), to $1.0$ (original distribution), to $1.5$ (high entropy). In Figure 1 we definitively confirm the existence of transcendence. Our ChessFormer 1000 (where the latter number refers to the maximum rating seen during training) and ChessFormer 1300 models are able to transcend to around 1500 rating at temperature $\tau$ equal to $0.001$. Interestingly, ChessFormer 1500 is unable to transcend at test time, a result we further analyze in Dataset Diversity.

To more deeply understand when and why transcendence occurs, we investigate two questions. (1) How does the reward function defined in Equation 2 shift with respect to low-temperature sampling? (2) Does transcendence rely on dataset diversity, as introduced theoretically in subsection 3.4?

**Lowering temperature increases rewards in expectation on specific states, leading to transcendence over the full game.**    When playing chess, a low-skilled player may play reasonably well

until they make a significant blunder at a key point in play. If these errors are idiosyncratic, averaging across many experts would have a denoising effect, leaving the best moves with higher probability. Therefore, low-temperature sampling would move probability mass towards better moves in specific play contexts. Without low-temperature sampling, the model would still put probability mass onto blunders. To gain intuition for this idea, we visualize it theoretically in Appendix C and empirically in Figure 2 and Appendix B. This hypothesis motivates our first research question in this section: Does low-temperature sampling improve the expected reward very much for just some specific key game states, or a little for many game states?

To formalize this notion, we first define a "favor" function, which captures the improvement in reward by following some new probability distribution over some baseline probability distribution. Our definition is inspired by the Performance Difference Lemma (PDL) [10] from Reinforcement Learning (RL), which establishes an equivalence between the change in performance from following some new policy (a probability distribution of actions given a state) over some old policy, and the expected value of the advantage function of the old policy sampled with respect to the new policy. In RL, the advantage function is defined as the difference between the value of taking a single action in a given state versus the expected value of following some policy distribution of actions in that state.

Here, we define the "favor" of $f'$ over $f$ in $x$ as the change in the reward function by comparing what $f$ would have done when following $f'$ for a given input $x$:

$$F(f',f;x) = \mathbb{E}_{x \sim d^{f'}, y \sim f'(\cdot|x)}[r(x,y)] - \mathbb{E}_{x \sim d^{f'}, y \sim f(\cdot|x)}[r(x,y)]. \qquad (4)$$

Where $d^f$ refers to the state visitation distribution [31] when following $f$ in a sequential setting—informally, this variable can be thought of the distribution of states seen when sampling from $f$ with a fixed transition function that takes in an input $x$, a output $y$, and outputs a next input $x$. Here, that transition function is given by the rules of chess and the opponent player. Given this favor function, we can now quantitatively explore the effects that lead to transcendence by setting the baseline $f$ to be the original imitation-learned probability distribution (temperature $\tau = 1$), and $f'$ as a low-temperature intervention on $f$ (e.g. temperature $\tau = 0$). We can empirically calculate the reward by using the evaluation function [22] of Stockfish, an expert neural reward function that Stockfish uses to calculate its next move. This reward function is a neural network trained to predict the probability of winning through a sigmoid on a linear combination of handcrafted expert heuristics, such as amount of material versus opponent material, and number of moves to a potential checkmate.

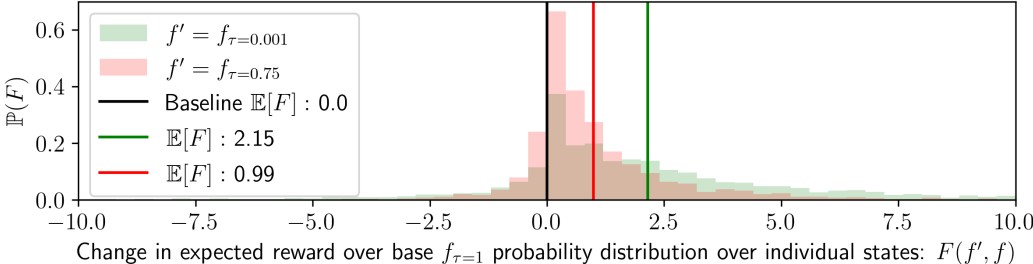

Figure 4: The favor probability distribution, or change in expected reward by setting temperature lower than $\tau = 1.0$. We plot the favor distribution across two different temperatures: setting $\tau = .75$ and $\tau = 0.001$ by running the Stockfish analysis engine across 100 total Chessformer 1000 games played at 0.001 temperature against Stockfish level 1 (as theoretically justified by PDL [10]). We calculate favor by sampling 100 counterfactual potential moves at $\tau = 1.0$ per actual move made at $\tau = 0.001$ to compute a baseline expected reward. In total, we gather an empirical probability distribution with $n = 382,000$ total samples per $\tau$ (38.2 moves on average per game). Note that we plot the distributions with transparency, so the brownish area is where the two overlap. We visualize several long-tail examples in Appendix B.

In Figure 4, we find that lowering the temperature has the effect of skewing the expected reward distribution to the right, especially for the green $\tau = 0.001$ distribution. This result implies that the model does not improve the expected reward by a small amount for many game states, but rather improves the expected reward by a relatively large amount for a few game states. Thus, $\tau = 0.001$ improves the expected reward (probability of winning) by an average of $\mathbf{2.15 \pm 0.17}\%$, but for some states, this expected improvement is over 5%. Note that the original temperature expected reward can

be thought of as a Dirac distribution centered at $0$. The above finding answers our research question in this section: Low-temperature sampling is able improves the expected reward by relatively large amounts for some specific game states, which is likely why the ChessFormer 1000 and 1300 model was able to achieve transcendence.

| Temperature | $\mathbb{E}[\mathbb{P}_\tau(\%)]$ | $\mathbb{E}[\mathbb{P}_\tau - \mathbb{P}_{1.0}]$ | Top 1 $Acc(\%)$ | Top 3 $Acc(\%)$ | Top 5 $Acc(\%)$ |
|---|---|---|---|---|---|
| $\tau = 0.001$ | **39.95±0.92** | **2.15±0.17** | **29.61±1.43** | **54.26±1.57** | **66.86±1.47** |
| $\tau = 0.75$ | 38.79±0.90 | 0.99±0.06 | 25.08±0.95 | 47.84±1.09 | 60.37±1.04 |
| $\tau = 1.0$ | 37.80±0.87 | 0±0 | 22.61±0.86 | 44.00±9.96 | 56.27±0.93 |

Table 1: Table of several statistics describing the relationship between reward at $\tau = 0$ vs. $\tau = 1$. In the first column, we display the expected reward across our dataset, which is $\mathbb{P}$ of winning calculated by Stockfish 16.1). In the second column, we display $F$, or the change in reward for the given temperature $\tau$ versus the baseline. In the last three columns we display the accuracy for the best moves ranked by Stockfish analysis run at a time cutoff of 1 second. Here, the top-$k$ accuracy is the percentage of games where the actual move sampled by the model was in the top-$k$ moves as ranked by Stockfish. We report 95% bootstrapped confidence intervals with 10K resamples.

In Table 1, we present the statistics of the favor function for different temperature values. From this table, we observe that as the temperature decreases, the top-$k$ accuracies monotonically increase, suggesting that the model becomes more consistent in selecting good moves. We also observe that although the model improves as temperature decreases, the probability of winning is still below $50\%$, meaning our model should tend to lose more games than it wins against Stockfish 1. This result matches with our results in Figure 1, as the rating of Stockfish 1 is also higher than the reported rating for $\tau = 0.001$ (1550 for Stockfish 1 vs $\sim 1450$ for Chessformer 1000). Overall, the analysis of the advantage statistics provides further evidence for the effectiveness of low-temperature sampling in inducing transcendence in chess models.

**Dataset diversity is essential for transcendence.** As we note in subsection 3.4, our theory requires dataset diversity as a necessary condition for enabling transcendence. Importantly, we find in Figure 1 that not all models are able to transcend. Unlike ChessFormer 1000 or 1300, the Chessformer 1500 fails to transcend. We hypothesize that this results is due to the fact that in the band of ratings from 1000 to 1500, diversity does not significantly increase. If so, a 1000 rated player can be thought of as a noisy 1500 rated player, but a 1500 rated player cannot be thought of as a noisy 2000 rated player. In this section we ask the following research question: Is diversity in data required for enabling transcendence?

In Figure 5, we explore this research question by quantifying dataset diversity through the normalized entropy on the action distribution $\mathcal{H}_f(Y|X) = \mathbb{E}_{y \sim f(y|x=X)}[-\log_2 f(y|x=X)]/\log_2 |\mathcal{Y}|$. To gain intuition for this metric, imagine the action distribution of moves taken for any given state. Entropy will be higher for more uniform action distributions, and lower for more deterministic, peaked action distributions. The average entropy of these action distributions can therefore serve as a measurement of the diversity of the dataset. We normalize this entropy to the range $[0,1]$ by dividing by the binary log of the number of legal moves: $\log_2 |\mathcal{Y}|$.

Importantly, we cannot calculate this normalized entropy for every state, as most states after move 16 in the midgame and before the engame are unique within the dataset and we therefore observe just a single action for thus states. Therefore our metric is limited in that it only considers opening moves, the beginning of the midgame, and the endgame. We consider only common states with greater than 100 actions by sampling $1,000,000$ games from each dataset. The average entropy confirm our hypothesis: The $< 1500$ cut off dataset has on average less diversity than the $< 1300$ dataset, which has is again less than the $< 1000$ dataset. This result suggests that Chessformer 1500 likely is not transcendent due to a lack of diversity in its dataset. If the entropy instead stayed constant for each dataset, it would imply that each had a similar level of diversity. In such a case, we would expect that ChessFormer 1500 likely would also transcend. Instead, as predicted, it is likely not transcendent due to a lack of diversity.

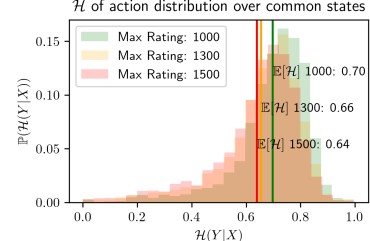

Figure 5: Action distribution diversity, as measured by the average normalized entropy over different chess rating dataset cutoffs with $n = 2681, 3037, 3169$ common states for ratings $1000, 1300, 1500$, respectively. These entropies are calculated directly from the empirical frequencies of our dataset, and are model-agnostic.

### 4.3 Additional Settings

**SQuADv2 Natural Language Temperature Denoising Experiment.** We extend our analysis to the Natural Language Processing domain by running experiments on the Stanford Question Answering Dataset (SQuAD 2.0). We tested the effects of temperature denoising on the performance of several large language models (LLMs) of varying sizes. The SQuAD task involves reading comprehension and question-answering based on Wikipedia articles, making it an ideal setting to evaluate the impact of denoising on language models. We measured the exact-match, semantic-match, and F1 scores of the model outputs at different temperatures. The results show that temperature denoising leads to improved performance, corroborating the findings of our chess experiments and providing broader validation of the underlying mechanism of temperature denoising in diverse domains.

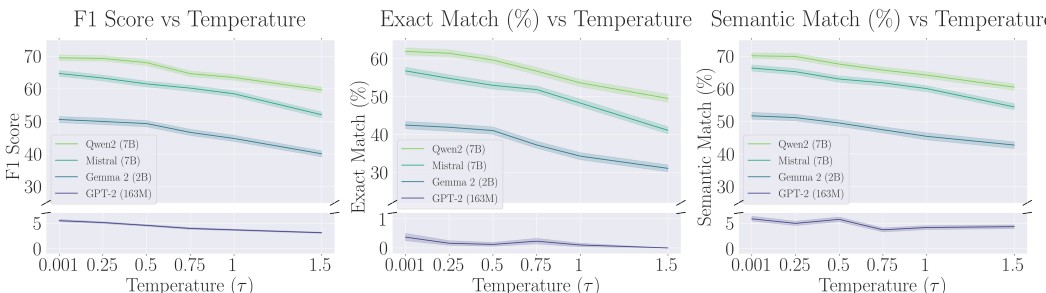

Figure 6: We evaluate several pretrained language models on the SQuADv2 Question-Answering reading comprehesion dataset, a task consisting of answering a question given some snippet from a Wikipedia article. We report F1, 'Exact Match', and 'Semantic Match' scores of several different language models of varying size from 163M parameters to 7B parameters, over several different temperatures. Semantic Match is calculated by using another LLM (llama3.1) to judge if two responses are equivalent, even if the exact strings slightly differ between the model output and the correct response. We also report 95% confidence intervals calculated through taking $\pm 1.96\sigma$.

**Toy Model Setting and Results.** In addition, we develop a toy theoretical model to further study when transcendence is possible. This model involves a classification task with Gaussian input data and linearly separable classes. Experts label the data with noisy versions of the ground truth separator. We trained a linear model on a dataset labeled by random experts and observed the test accuracy for different temperature settings. The synthetic experiments demonstrated that transcendence occurs when expert diversity is high and temperature is low, aligning with our theoretical and empirical analysis in the chess domain.

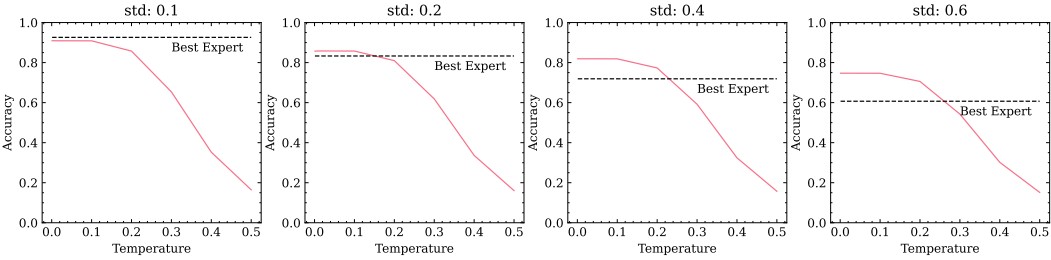

Figure 7: Toy model for demonstrating transcendence. Input data is $d$-dimensional Gaussian, with $d = 100$. Output is classification with 10 classes. Ground-truth is generated by a linear function, i.e. $y = \mathrm{argmax}_i W_i^\star x$ for some $W^* \in \mathbb{R}^{10 \times d}$. We sample $k$ experts, with $k = 5$, to label the data, where the labels of each expert are generated by some $W \in \mathbb{R}^{10 \times d}$ s.t. $W = W^* + \xi$, where $\xi_{i,j} \sim \mathcal{N}(0, \sigma^2)$, for some standard deviation $\sigma$. Namely, each expert labels the data with a noisy version of the ground truth separator, with noise std $\sigma$. We then train a linear model on a dataset with $10K$ examples, where each example is labeled by a random expert. We plot the test accuracy, measured by the probability assigned to the correct class, for different choices of temperature, and compare to the best expert.

# 5 Related Work

**Chess and AI.** Chess has been motivating AI research since the field began. In 1950, before anyone had used the term "artificial intelligence", automated chess were explored by both Claude Shannon [26] and Alan Turing [32]. Arguably, this history goes back even further: the famed "mechanical turk" of the 18th century was a fraudulently automated chess player. These centuries of mechanical ambitions were finally realized in 1997, when world champion Garry Kasparov was defeated by IBM's Deep Blue [3]. Since then, chess program developers have drawn on neural approaches, with the RL-based convolutional network AlphaZero [27] far surpassing prior world champion engines such as Stockfish [25].Our chess model testbed is inspired by a number of existing approaches, including other models trained on lichess data [18], and other transformer-based sequential chess agents [23, 5].

**Diversity beats Strength.** Another historical thread in AI research is the strength of diverse learners. Long since the development of ensemble methods that exploit learner diversity—including bagging [1], boosting [6], and model averaging [19]—researchers have continued to articulate this insight across settings. Similar to our chess setting, a diverse team of go playing agents have been proven and empirically shown to outperform solitary agents [9] and homogeneous teams [28], even when the alternative models individually outperform the diverse team members [17]. We draw a connection to this deep literature through our theory, which shows that imitation learning objective and then performing low-temperature sampling subtly implies the same principle of majority voting. Teacher diversity has also been explored in the machine learning literature. One related method is ensemble distillation [16], in which a model is trained with an additional objective to match a variety of weaker teacher models. Closer to our setting, ensemble self-training approaches [24] train a learner directly on the labels produced by varied teachers. Large language models supervised by smaller or less trained models are said to exhibit "weak to strong generalization" [2]. Overall, evidence continues to accrue that the general phenomenon we address is pervasive: that is, models can substantially improve over the experts that generate their training data.

**Offline Reinforcement Learning.** Our work also draws connections to the Offline Reinforcement Learning [14] setting, where one attempts to learn a new policy $\pi$ that improves upon a fixed dataset generated by some behavior policy $\pi_\beta$. However, our setting of imitation learning differs substantially from this literature, as we do not explicitly train our model on a RL objective that attempts to improve upon the dataset. Importantly, such an objective oftentimes introduces training instabilities [15] and also assumes reward labels. We defer a more extended discussion of related work to Appendix D.

# 6 Discussion and Future Work

This paper introduces the concept of transcendence. Our theoretical analysis shows that low-temperature sampling is key to achieving transcendence by denoising expert biases and consolidating diverse knowledge. We validate our findings empirically by training several chess models which, under low-temperature sampling, surpass the performance of the players who produced their training data, as well as further experiments in natural language question-answering and toy Gaussian models. We additionally highlight the necessity of dataset diversity for transcendence, emphasizing the role of varied expert perspectives.

**Limitations.** While our work provides a strong foundation for understanding and achieving transcendence in generative models, several avenues for future research remain. Future work may investigate transcendence and its causes in domains and contexts beyond chess, such as natural language processing, computer vision, and text-to-video, to understand the generalizability of our findings. Additionally, our theoretical framework assumes that game conditions at test time match those seen during training; in order to extend our findings to cases of composition or reasoning, we must forego this assumption.

**Future Work.** Future work could also explore the practical implementations of transcendence, and ethical considerations in the broader context of deployed generative models. Ultimately, our findings lay the groundwork for leveraging generative models to not only match but exceed human expertise across diverse applications, pushing the theoretical boundaries of what generative models can achieve.

**Broader Impact.** The possibility of "superintelligent" AGI has recently fueled many speculative hopes and fears. It is therefore possible that our work will be cited by concerned communities as evidence of a threat, but we would highlight that the denoising effect addressed in this paper does not offer any evidence for a model being able to produce novel solutions that a human expert would be incapable of devising. In particular, we do not present evidence that low temperature sampling leads to novel abstract reasoning, but just denoising of errors.

## Acknowledgements

Sham Kakade acknowledges this work has been made possible in part by a gift from the Chan Zuckerberg Initiative Foundation to establish the Kempner Institute for the Study of Natural and Artificial Intelligence; support from the Office of Naval Research under award N00014-22-1-2377, and the National Science Foundation Grant under award #IIS 2229881.

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

# A  Proofs

Here we prove Proposition 1, where *transcendence*cannot occur by purely using imitation learning in our setting where all experts are sampled uniformly across the input distribution.

*Proof.* From linearity of the expectation

$$R_{p_{\text{test}}}(\hat{f}) = \mathbb{E}_{x \sim p_{\text{test}}}\left[r_x(\overline{f})\right]$$

$$= \mathbb{E}_{x \sim p_{\text{test}}}\left[\frac{1}{k}\sum_{i=1}^{k}r_x(f_i)\right] = \frac{1}{k}\sum_{i=1}^{k}R_{p_{\text{test}}}(f_i) \leq \max_i R_{p_{\text{test}}}(f_i)$$

$\square$

We now give the proof of Proposition 2 that if the arg-max prediction is better than the best expert, then transcendence is possible with low-temperature sampling.

*Proof.* Observe that for all $q$, it holds that $\lim_{\tau \to 0}\text{softmax}(q;\tau) = \text{argmax}(q)$. Therefore, for all $x$

$$\lim_{\tau \to 0}r_x(\hat{f}_\tau) = \lim_{\tau \to 0}\sum_y r(x,y) \cdot \hat{f}_\tau(y|x) = \sum_y r(x,y)\hat{f}_{\max}(y|x) = r_x(\hat{f}_{\max})$$

and so,

$$\lim_{\tau \to 0}R_{p_{\text{test}}}(\hat{f}_\tau) = \lim_{\tau \to 0}\mathbb{E}_{x \sim p_{\text{test}}}\left[r_x(\hat{f}_\tau)\right]$$

$$= \mathbb{E}_{x \sim p_{\text{test}}}\left[\lim_{\tau \to 0}r_x(\hat{f}_\tau)\right] = \mathbb{E}_{x \sim p_{\text{test}}}r_x(\hat{f}_{\max}) = R_{p_{\text{test}}}(\hat{f}_{\max})$$

Therefore, the required immediately follows. $\square$

To prove Proposition 3.3, we directly use the result in Proposition 2.

*Proof.* Notice that for this expert, $\text{argmax}(f(\cdot|x)) = f^*(y|x)$, which achieves higher reward compared to $f$. Therefore, Theorem 2 implies that we achieve transcendence in the setting where all the data is generated by a single expert $f$. $\square$

Next, we give the proof for that low-temperature sampling can be thought of as performing *majority vote* [1, 6] between the experts:

**Proposition 5.** *Let $\mathbf{z} = [z_1, z_2, ..., z_n]$ be a vector and $\tau > 0$ be a temperature parameter. Define the softmax function as*

$$\sigma_\tau(z_i) = \frac{e^{z_i/\tau}}{\sum_{j=1}^{n}e^{z_j/\tau}}.$$

*Then, as $\tau \to 0^+$, the limit of the softmax function is given by*

$$\lim_{\tau \to 0^+}\sigma_\tau(z_i) = \begin{cases} \dfrac{1}{k}, & \text{if } z_i = z_{\max}, \\ 0, & \text{otherwise}, \end{cases}$$

*where $z_{\max} = \max_{1 \leq j \leq n}z_j$, and $k$ is the number of indices $i$ such that $z_i = z_{\max}$.*

*Proof.* Let $z_{\max} = \max_{1 \leq j \leq n}z_j$, and define the set

$$S = \{i \,|\, z_i = z_{\max}\},$$

with cardinality $k = |S|$.

For each $i$, let

$$\Delta_i = z_i - z_{\max} \leq 0.$$

Then the softmax function becomes

$$\sigma_\tau(z_i) = \frac{e^{(z_{\max}+\Delta_i)/\tau}}{\sum_{j=1}^n e^{(z_{\max}+\Delta_j)/\tau}} = \frac{e^{\Delta_i/\tau}}{\sum_{j=1}^n e^{\Delta_j/\tau}},$$

since $e^{z_{\max}/\tau}$ cancels out in the numerator and denominator.

We analyze the behavior of the terms as $\tau \to 0^+$.

For $i \in S$ we have $\Delta_i = 0$ and so:

$$e^{\Delta_i/\tau} = e^0 = 1.$$

For $i \notin S$ we have $\Delta_i < 0$ so

$$\lim_{\tau \to 0} e^{\Delta_i/\tau} = 0$$

Therefore, the denominator simplifies to

$$\lim_{\tau \to 0^+} \sum_{j=1}^n e^{\Delta_j/\tau} = \sum_{j \in S} \lim_{\tau \to 0^+} e^{\Delta_j/\tau} + \sum_{j \notin S} \lim_{\tau \to 0^+} e^{\Delta_j/\tau} = \sum_{j \in S} 1 + \sum_{j \notin S} 0 = k.$$

Similarly, the numerator becomes

$$\lim_{\tau \to 0^+} e^{\Delta_i/\tau} = \begin{cases} 1, & \text{if } i \in S, \\ 0, & \text{if } i \notin S. \end{cases}$$

Thus, for each $i$,

$$\lim_{\tau \to 0^+} \sigma_\tau(z_i) = \frac{\lim_{\tau \to 0^+} e^{\Delta_i/\tau}}{\lim_{\tau \to 0^+} \sum_{j=1}^n e^{\Delta_j/\tau}} = \begin{cases} \frac{1}{k}, & \text{if } i \in S, \\ 0, & \text{if } i \notin S. \end{cases}$$

This concludes the proof. $\qquad\square$

Finally, we give the proof of Proposition 4, or the statement that transcendence can occur from multiple experts if the test distribution $p_{\text{test}}$ is spread across multiple disjoing subsets of $\mathcal{X}_i$.

*Proof.* In this case, observe that for all $i$

$$R_{p_{\text{test}}}(f_i) = p_{\text{test}}(\mathcal{X}_i) \cdot \mathbb{E}_{x \sim p_{\text{test}}|\mathcal{X}_i} r_x(f^*) + p_{\text{test}}(\mathcal{X} \setminus \mathcal{X}_i) \cdot \mathbb{E}_{x \sim \overline{p}|\mathcal{X} \setminus \mathcal{X}_i} \left[ \mathbb{E}_{y \sim \text{Uni}(\mathcal{Y})} r(x,y) \right]$$
$$< R_{p_{\text{test}}}(f^*)$$

Therefore, we get that for all $x$

$$\hat{f}(y|x) = \frac{1}{k} \sum_{j=1}^k f_j(y|x) = \frac{k-1}{k} \cdot \frac{1}{|\mathcal{Y}|} + \frac{1}{k} f^*(y|x) = \frac{k-1}{k \cdot |\mathcal{Y}|} + \frac{1}{k|Y_x^*|} \cdot \mathbf{1}_{y \in Y_x^*}$$

Thus, we get $f_{\max} = f^*$, and the required follows from Proposition 2.

$\qquad\square$

# B  Additional Denoising Visualizations

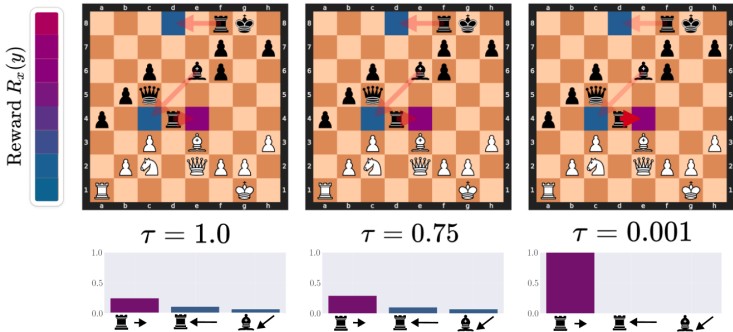

Figure 8: An example of where denoising helps black find the only correct move. White has pinned the black rook to the Queen: any move where the rook does not move to e4 results in a heavy loss of material. As $\tau$ decreasses, the expected reward increases substantially and converges onto the correct move.

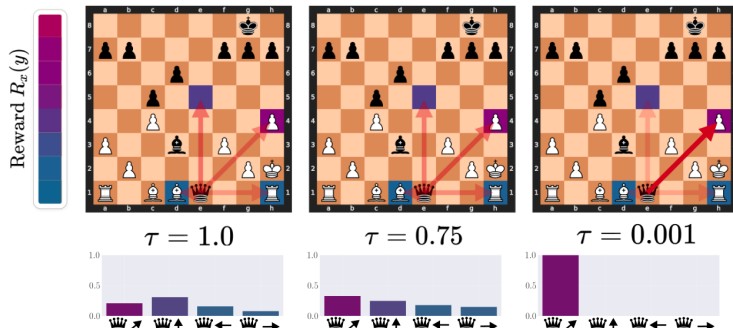

Figure 9: Another example where denoising helps avoid errors. Moving the queen to either d1 or h1 takes a bishop or rook, respectively, but loses the queen in the following turn. While queen to e5 does not put the queen in immediate danger, it allows white to push the pawn on f3 to d3, where it threatens the queen and is protected by the bishop on c1. The queen then must move out of danger, losing its opportunity to take the free pawn on h4 and giving white valuable space towards the center of the board. As $\tau$ decreases, the expected reward converges to the move queen to d4, taking the pawn and checking the black king.

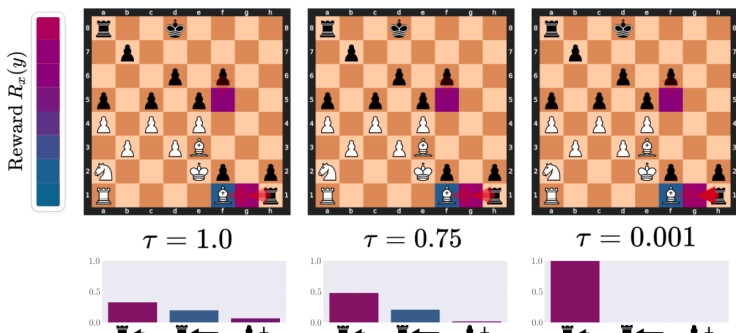

Figure 10: In this setup, a higher temperature shows two plausible moves for the black rook: g1 or f1. As the temperature decreases, the expected reward converges to g1. If the black rook were to move to f1, the white rook would take the black rook, blocking the black pawn on f2 from promoting and protecting the promotion square from the h2 pawn. If the rook were to move to g1, on the other hand, it would open the promotion square from the h2 pawn without being at any immediate risk. If white responded by moving its bishop to g2, protecting the promotion squares from both of the advanced black pawns, black could respond by taking the rook on a1, gaining significant material.

## C   Intuition of low temperature sampling inducing transcendence

To build intuition for the primary mechanism of transcendence that we explore in this paper, we give the following toy progression of distributions in order to clearly illustrate how low-temperature sampling can induce transcendence through majority voting. Here, the middle purple action represent the correct, high-reward output, whilst the left and right actions are low-reward bad outputs. We plot the probability of each output as a label on the x axis.

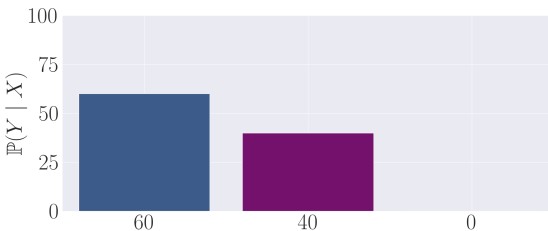

Figure 11: The first expert output distribution. Although it puts non-negligible mass on the purple, high-reward action, it still samples a low-reward action the majority of the time.

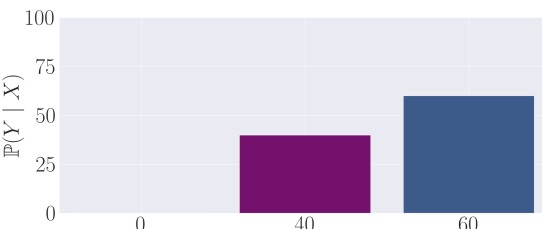

Figure 12: The second expert output distribution. Symmetric to to the first expert, it also puts non-negligible mass on the purple, high-reward action. However, it samples a low-reward action the majority of the time on the right.

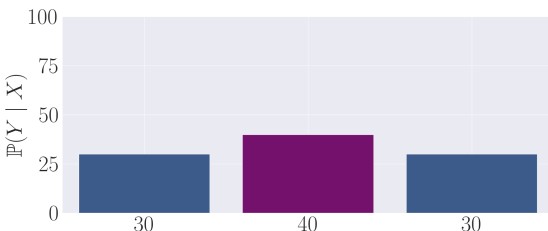

Figure 13: By taking the average of the first and second expert, we observe that this distribution now puts the majority of mass onto the correct action.

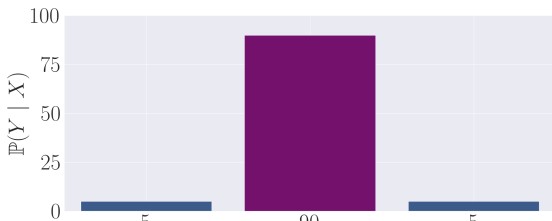

Figure 14: Finally, by setting temperature $\tau$ to be $< 1$, more weight is shifted towards the high probability action, leading to a gain in the expected reward.

# D    Further Related Work

## D.1    Label Disagreement

Label disagreement in training data, in particular, can improve models in practice. Xie et al. [36] empirically show that adding random noise to teacher-generated labels can improve a student model. Uma et al. [33] even survey the literature on human interannotator disagreement and find a trend of improvements when models are trained on the full set of disagreeing labels rather than on majority vote labels or only on data where labelers agree. Our theoretical claims build on these findings by making the point that the learner can even improve on these original diverse labelers.

## D.2    Offline Reinforcement Learning

Although most Offline Reinforcement Learning algorithms train on an RL objective, perhaps most similar to our work is Decision Transformer [4] and Trajectory Transformer [8]: prior models trained on just the sequence prediction of trajectories. Most notably, Decision Transformer also finds an alternative form of *transcendence* than the one explored in this paper: by conditioning the trained transformer by the performance of the trajectory, at inference time they can then prompt the model to perform better than the best trajectory seen during training. This remains another promising direction to explore transcendence under.

Interestingly, an analogue to low-temperature sampling also has been noticed and exploited by Reinforcement Learning practitioners in the context of *off-policy learning*, where a different exploration policy $\pi_E$ is used than the final learned target policy $\pi_T$. Oftentimes $\pi_T$ will just be set to a greedy version of $\pi_E$ [21], such as choosing $\pi_T$ to take the argmax action of $\pi_E$, which we note is directly equivalent to setting temperature to 0.

# E  Training Details

We give a full list of the hyperparameters we used for training here. Note that we largely follow the same hyperparameter set as [37], but lower the batch size to $125K$ as we found training to still be stable ta this level. We also release our code openly to support further research into transcendence, which was built off the wonderful work done by Karvonen [12] and Karpathy [11].

|  | Hyperparameter | Value |
|---|---|---|
| ChessFormer | Optimizer | AdamW [13] |
| | Activation Function | ReLU |
| | Mini-batch size | 125K tokens |
| | Gradient Accumulation Steps | 1 |
| | Transformer num. layers | 16 |
| | Transformer num. heads | 8 |
| | Transformer embedding dim. | 512 |
| | Dropout | 0.0 |
| | Learning Rate | 3e-4 |
| | Number of gradient steps | 100K |
| | Weight Decay | 0.1 |
| | Critic hidden layers | 3 |
| | Adam $\beta_1$ | 0.90 |
| | Adam $\beta_2$ | 0.95 |
| | Gradient Clip | 1.0 |
| | Cosine Learning Rate | True |
| | Warmup Iterations | 2000 |
| | Minimum Learning Rate | 3e-5 |
| | Learning Rate Deacy Iterations | 400K |
| | Tensor datatype | bfloat16 |

Table 2: Hyperparameters for our ChessFormer model.

# F    Compute Resources

We train all of our models on the Nvidia H100 80GB GPU. To train one of our models takes around 6 to 12 hours.

## G    Full t-SNE

We visualize the full t-SNE here, coloring by the reward of the game. We see that the model has learned some representation of the reward, with high absolute reward states being more likely to be near each other in the latent space. This also points towards evidence that the model has learned some sort equivariant representation of the player identity, as the region of symmetric high reward states indicate. Note that reward is not directly given to the model during training.

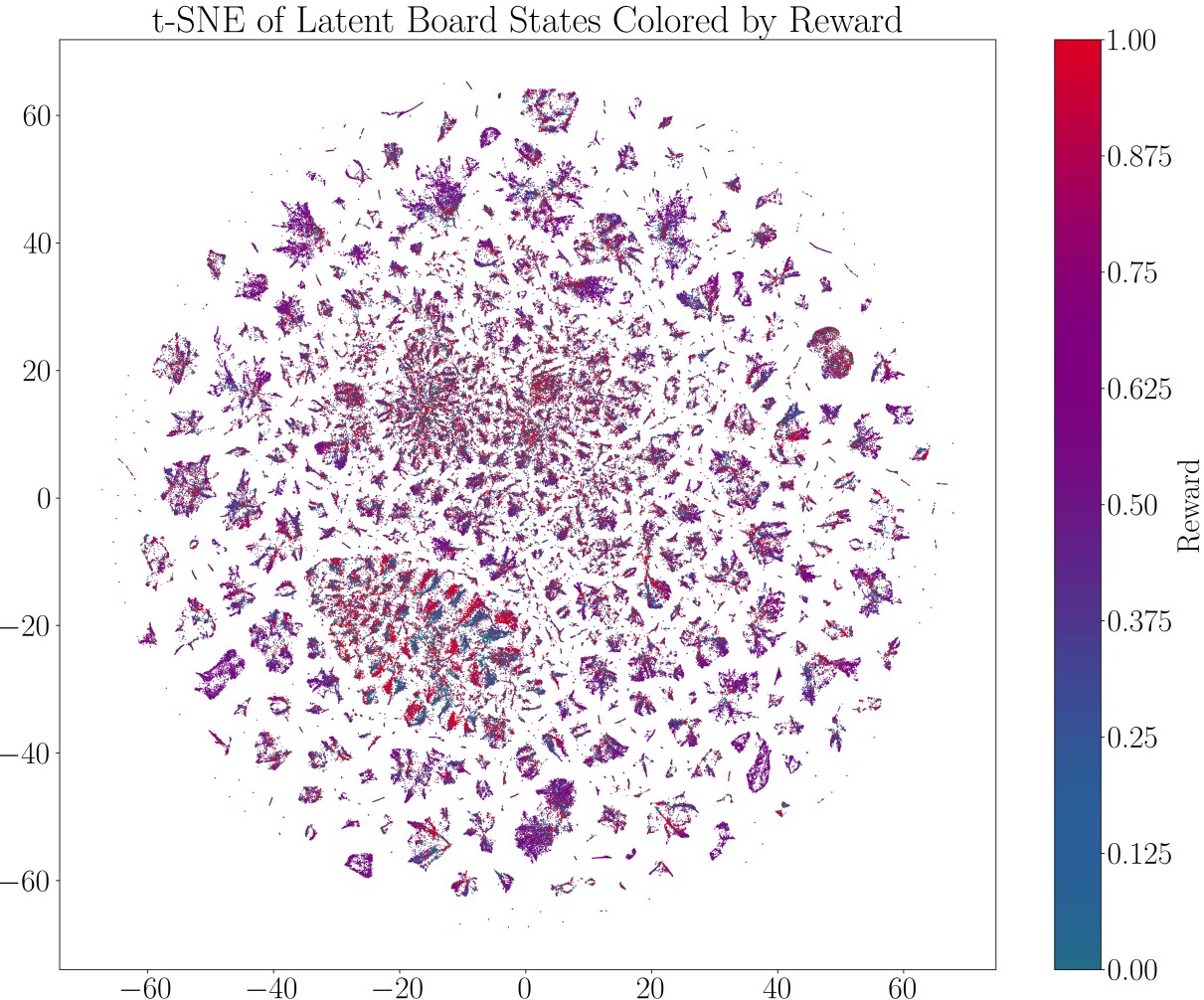

We visualize the same t-SNE, but this time coloring by game length rather than reward. We see that games with high reward tend to be longer, which makes logical sense as the result of the game will tend to be clearer as the game proogresses.

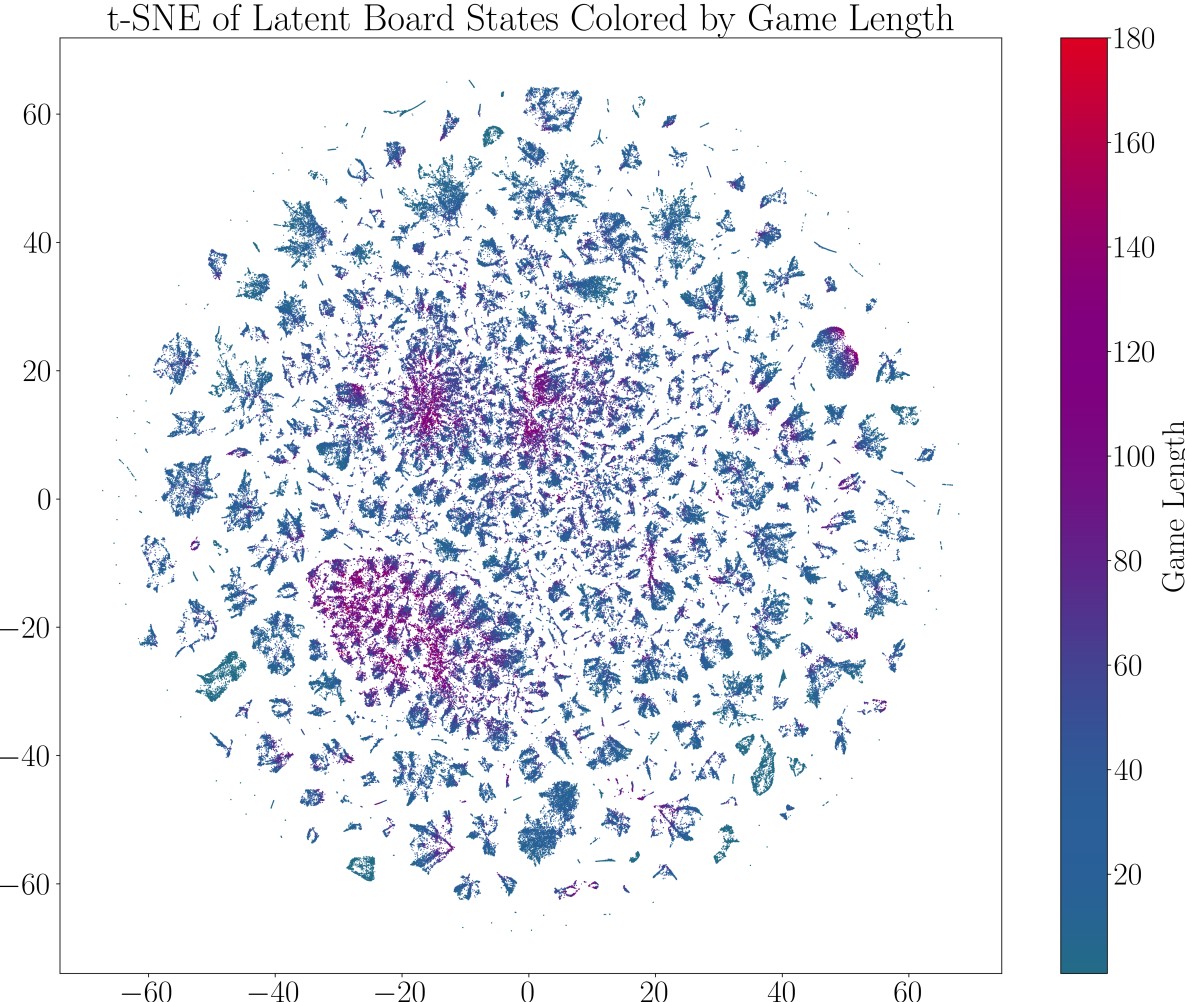

