# OpenReview forum: "Transcendence: Generative Models Can Outperform The Experts That Train Them"
_NeurIPS.cc/2024/Conference — NeurIPS 2024 poster_

### Official Review · Reviewer_np7r · 2024-07-06

**Soundness:** 3
**Presentation:** 3
**Contribution:** 3
**Rating:** 7
**Confidence:** 3

**Summary:**

1. Definition of Transcendence: The paper defines transcendence as a phenomenon where a generative model, trained on data from human experts, achieves capabilities that surpass the abilities of those experts.

2. Theoretical Framework: The authors provide a theoretical analysis of the conditions under which transcendence can occur. They prove that low-temperature sampling is necessary for transcendence in their setting.

3. Chess as a Testbed: The researchers use chess as a domain to empirically test their theories, training transformer models (called "ChessFormers") on datasets of human chess games with various skill levels.

4. Main Findings:
   - Low-temperature sampling enables transcendence: Models trained on games from players rated up to 1000 and 1300 were able to achieve ratings above 1500 when using low-temperature sampling.
   - Dataset diversity is crucial: The model trained on games up to 1500 rating did not achieve transcendence, likely due to less diversity in the dataset.
   - Denoising effect: Low-temperature sampling helps by shifting probability mass towards better moves in key situations.

5. Theoretical Insights: The paper draws connections between their findings and existing literature on model ensembling and the "wisdom of the crowd" effect.

**Strengths:**

The paper studies an interesting questions theoretically and shows the insights relate on a nicely accessible empirical problem namely chess. One strength of the paper are also the simple theoretical arguments presented, which seem relevant in non-trivial settings. Paper is quite well written (one could improve here quite a bit) and results nicely presented, although the paper seems quite rushed.

**Weaknesses:**

I think the theoretical section could be improved by integrating the proofs, when given the extra page if accepted. Also I wouldn’t call the theoretical insights „theorems“ but rather propositions.
I think the paper would greatly benefit from a toy model which allows to study when transcendence is possible, theoretically and practically. I think the major weakness of the paper is the missing understanding when transcendence can occur and when it can’t.

Didn’t have much time to review this paper unfortunately, will study it in more detail in the next weeks and consider raising my score, after the authors feedback.

**Questions:**

See weaknesses

Typo in line 100 we make assume
Typo in line 123 outputs a correct but noisy prediction(s).
Reference in line 201 - reference to the current section
Typo in line 237: to the right?
Figure 4, improve legend and generally readability/visualizations of the Figure, E[F] two times, not clear what the difference is
Typo in line 249 This is matches
Typo in line 265 We, we
Reference Theorem line 142

**Limitations:**

Looks fine, would be great to  discuss future research directions.

---

> ### Author Rebuttal · Authors · 2024-08-06
>
> > We appreciate the reviewer's insightful questions and valuable critiques. To summarize, we have addressed concerns about a toy theoretical model, and typographical errors. We trust that our responses below will provide a more comprehensive understanding of our research and its implications.
>
> ---
> I think the theoretical section could be improved by integrating the proofs, when given the extra page if accepted. Also I wouldn’t call the theoretical insights „theorems“ but rather propositions.
>
> > We agree with the reviewer's sentiment, and will rename the theorems to propositions instead, and will integrate the proofs given the extra page if accepted.
>
> I think the paper would greatly benefit from a toy model which allows to study when transcendence is possible, theoretically and practically.
>
> > We agree that the paper can benefit from a toy model for studying transcendence. Following your suggestion, we introduce the following toy model, demonstrating transcendence in the case of Gaussian data and linearly separable classes. Specifically, our input data is $d$-dimensional Gaussian, and the target is classification with $n$ classes. The ground-truth is generated by a linear function, i.e. $y = \arg \max_i W_i^\star x$ for some $W^* \in \mathbb{R}^{n \times d}$. We sample $k$ experts to label the data, where the labels of each expert are generated by some $W \in \mathbb{R}^{n \times d}$ s.t. $W = W^* + \xi$, where $\xi_{i,j} \sim \mathcal{N}(0, \sigma^2)$, for some standard deviation $\sigma$. Namely, each expert labels the data with a noisy version of the ground truth separator, with noise std $\sigma$. We then train a linear model on a dataset with $m$ examples, where each example is labeled by a random expert. We observe the test accuracy, measured by the probability assigned to the correct class, for different choices of temperature, and compare to the best expert. As can be seen in our synthetic experiments (**in the attached rebuttal PDF**), we are able to observe transcendence in exactly the same situation as in our chess setting, i.e. - when the diversity of the experts is high (high std) and the temperature is low. This also aligns with our theoretical analysis.
>
> I think the major weakness of the paper is the missing understanding when transcendence can occur and when it can’t.
>
> > We would like to clarify that our paper provides a thorough analysis of the conditions necessary for transcendence both theoretically and empirically. Specifically, we detail in Section 3 the theoretical conditions for transcendence, emphasizing the necessity of low-temperature sampling (Section 3.1) and dataset diversity (Section 3.4). Our empirical results in Section 4 further validate these theoretical insights.
> > * Theoretical Analysis (Section 3): We analyze the necessity and sufficiency of low-temperature sampling and demonstrate that without it, transcendence cannot be achieved (Theorem 1). We also explore conditions involving multiple experts and the role of data diversity.
> > * Empirical Validation (Section 4.2): Our experiments confirm that low-temperature sampling enables transcendence to occur on high-diversity datasets by denoising errors (Figure 2 and Figure 4). We show that models trained on less diverse datasets, such as ChessFormer 1500, fail to transcend, highlighting the importance of dataset diversity (Figure 5).
> > * Toy Model: We note that our new toy model (see Rebuttal Summary) also cleanly demonstrates that transcendence is possible only in cases where the expert diversity is high, and the sampling temperature is low.
>
>
> See weaknesses
>
> * Typo in line 100 we make assume
> * Typo in line 123 outputs a correct but noisy prediction(s).
> * Reference in line 201 - reference to the current section
> * Typo in line 237: to the right?
> * Figure 4, improve legend and generally readability/visualizations of the Figure, E[F] two times, not clear what the difference is
> * Typo in line 249 This is matches
> * Typo in line 265 We, we
> * Reference Theorem line 142
>
> > We apologize for the typographical errors and unclear references you noted. We have corrected the following issues in our local version and will upload the revisions in the final version of our paper if it is accepted.
>
> > In response to the Figure 4 concerns, we would like to clarify that it presents the expected reward distributions at two different temperatures, specifically $\tau=0.75$ and $\tau=0.001$. The two instances of E[F] represent the expected reward under these two temperature settings. We will revise the figure legend and accompanying text to make this distinction clearer."
>
> Looks fine, would be great to discuss future research directions.
>
> > We would like to kindly clarify that we do lay out future research directions in Section 6, Discussion and Future Work. Some possible future work may include seeking other causes of Transcendence, expanding our analysis to more domains such as NLP, CV, and RL, and finally potentially using Transcendence in an iterative fashion to achieve a stable RL-like algorithm without learning a critic function but purely through Imitation Learning. All of these directions would be quite interesting to pursue.
>
> ---
>
> > Given these clarifications and the thorough addressing of your concerns, we hope the reviewer may consider raising their score. We are confident that our detailed response and amendments underscore the rigor and scalability of our work, potentially driving future insights into generative models. We trust that these responses adequately address your concerns. Your insight has been highly constructive and we appreciate your time in this review process. We hope to hear from you soon.

---

> > ### Comment · Reviewer_np7r · 2024-08-08
> > **Thank you!**
> >
> > Thank you for the additional results and analyses provided. After also studying the other reviews and responses, I will raise my score accordingly. While I think the insights aren't of very surprising nature, the paper provides clear evidence and theory for a interesting question/phenomenon.

---

> > > ### Author Response · Authors · 2024-08-08
> > > **Thank you!**
> > >
> > > Thank you for going through the other results and reviews, and for considering our responses to your initial comments. Your feedback has been helpful in strengthening our work by removing typos and advising on better flow for the theory section, and we are glad to hear that our paper's clarity and contribution to this interesting phenomenon were well-received. Thank you for your thoughtful review and for raising your score accordingly.

---

### Official Review · Reviewer_Se7B · 2024-07-11

**Soundness:** 3
**Presentation:** 3
**Contribution:** 3
**Rating:** 7
**Confidence:** 4

**Summary:**

In this work the authors focus on formalizing and investigating the concept of *trascendence*, i.e. the capacity of a model to outperform the experts it was trained on. To do so, they train transformer-based models for chess limiting experts used for training to a certain maximal ranking. In this setup they postulate and prove a connection of trascendence to low temperature scaling and validate it empirically. Then they continue their analysis by showing that the gain in performance due to low temperature scaling is to be attributed to large improvements over a relatively small subset of states. Finally, a link between trascendence and dataset diversity is drawn.

**Strengths:**

- The topic of the work is relevant, and the paper provides interesting insights with solid empirical validation.
- The choice of chess as an experimental setting is very smart for this study. This setup also allows the authors to perform targeted ablations and validate specific claims.
- The experiments section is clear, with convincing results, and it is easy for the reader to find and understand the evidence that backs up specific claims in the paper.
- The paper is well-structured and well-written.

**Weaknesses:**

- Chess is the only experimental setting considered. While the experiments are thorough in this setting, validating the claims in the paper on multiple and diverse experimental settings would be stronger evidence.
- Theorem 2, basically easily follows from the assumption that $\hat{f}_{max}$  is better than the best expert. Since it is so central, can the authors elaborate more on the assumption that $\hat{f} _{max}$  is better than the best expert? Is it always realistic to assume?
- The connection between majority voting and low temperature sampling (line 116) should be more formally proved, rather than left to intuition. One could substitute the expression for $\hat{f} =\frac{1}{k} \sum_{ i=1}^k f_i$ in $\text{softmax}(\hat{f}(\cdot|x); \tau)$ and show that for low $\tau$ (e.g. taking the $\lim _{\tau \rightarrow 0}$) it results in majority voting. In addition, I am not sure how much this connection is precise. In fact, looking at the Appendix C, doing majority voting of the single experts would not lead to the same outcome as low temperature sampling. It seems more precise to say that low temperature sampling results in a sharp (i.e. low-entropy) distribution with its peak corresponding to the action that is put more mass on by the consensus of experts.
- To my understanding, in Section 3.4 the mathematical definition of the i-th expert, that performs well on the subset $\mathcal{X}_i$, should rather be $f_i(y|x) = \biggl( \frac{\delta(y \in Y^\star_x) \delta(x \in \mathcal{X}_i)} {|Y^\star_x|} + \frac{\delta(x \notin \mathcal{X}_i)} {|\mathcal{Y}|} \biggr)$.  Am I missing something?
- In key parts of the work there are typos/inaccuracies (see above and Questions section), which hamper the clarity of the work and confuse the reader.

**Questions:**

- In Eq.4 there appears to be a typo, since the two terms subtracted on the right-hand-side are exactly the same. In my understanding it should be $y \sim f(.|x)$ in the second expectation.
- In line 237 it seems that "left" should be replaced with "right".

**Limitations:**

The authors are throughout the paper clear in the assumptions that define the scope of their claims, hence implicitly outlining some limitations of the study. Limitations are also explicitly addressed in the last section.

---

> ### Author Rebuttal · Authors · 2024-08-06
>
> > We thank the reviewer for their perceptive questions and feedback on the choice of chess as a setting and clarifications about the theory. In the responses following, we have sought to address each point in detail. We hope our clarifications will elucidate a better understanding of our approach and its potential to influence this research field.
> ---
> - Chess is the only experimental setting..
>
> > While chess provides a well-understood and controlled environment for initial validation, we agree that demonstrating our findings across diverse domains would strengthen our claims. Thus, we have run a new preliminary experiment on the Stanford Question Answering Dataset ([SQuAD 2.0](https://rajpurkar.github.io/SQuAD-explorer/)), testing the effects of temperature denoising performance on several LLMs of various sizes. The SQuAD task is a reading-comprehension question-answering task on Wikipedia articles. This setting highlights how denoising improves performance in language models as these transformers begin to make fewer errors in language generation. Empirically, we measure the exact-match and F1 score on the output generations of the LLM at different temperatures. Other prior work has also found ensemble models outperform humans on this task [1,2,3]. As we explain within our paper, ensembling and majority-voting can be thought of as another form of temperature denoising, as these methods all exploit the same underlying property of learner diversity. This expansion provides a broader validation of better performance through low-temperature denoising. We include these results in the rebuttal response.
>
> > In addition, we cite in Appendix D ("Further Related Work") that several prior works have also found that pure imitation learning can lead to generative models that outperform the dataset in both the domains of Reinforcement Learning [4] (Atari games) and Natural Language Processing [5] (next-token prediction task), although they do not elucidate the mechanisms under why such phenomenon may occur. In our work, we give one such mechanism of transcendence through temperature denoising and support it with rigorous theoretical and empirical evidence.
>
> > [1] Li, Shilun, et al. "Ensemble ALBERT on SQuAD 2.0." 19 Oct. 2021, arxiv.org/abs/2110.09665
>
> > [2] Zhang, Zhuosheng, et al. "Retrospective Reader for Machine Reading Comprehension." 11 Dec. 2020, arxiv.org/abs/2001.09694v4.
>
> > [3] Rajpurkar, Pranav, et al. "SQuAD Explorer." Stanford Question Answering Dataset.
>
> > [4] Figure 4, Section 5.2. Chen, Lili, et al. "Decision Transformer: Reinforcement Learning via Sequence Modeling." 24 June 2021, arxiv.org/abs/2106.01345.
>
> > [5] Shlegeris, B., Roger, F., and Chan, L. Language models seem to be much better than humans at next-token prediction. Alignment Forum, 2022.
>
> - Theorem 2, basically easily follows..
>
> > Thank you for asking for further clarification. We kindly refer you to Section 3.2, where we elaborated further on the assumption that ${f_{max}}$ is better than the best expert. We discuss scenarios where this assumption holds true, particularly in contexts where the model benefits from aggregating diverse expert knowledge, leading to performance that exceeds individual experts. To be clear, this is not always realistic to assume. For instance, in the paper, we demonstrate that one cannot assume that $\hat{f}_{max}$ is always better than the best expert in the 1500-rating chess example. Due to the lower data diversity, aggregating the different experts does not result in better performance than the best expert. Another example is given in the new toy model that we propose (see the Rebuttal Summary), where training on data generated by non-diverse experts (low std) does not result in transcendence.
>
> - The connection between majority voting and low temperature sampling..
>
> > We will give here a short proof for this result, and add a more detailed proof to the final revision of the paper. Let $z$ be the vector of outputs, i.e. $z = \hat{f}(\cdot|x)$, and denote $s = \text{softmax}(z|\tau)$, where $\tau$ is the temperature. Let $a = \max z$ be the maximal value in $z$, or the value of the logit(s) for the majority vote. Let $i$ be some coordinate of $z$ that is strictly smaller than $a$, namely $z_i < a$. Then, observe that $s_i = \frac{\exp(z_i/\tau)}{\sum_{j}\exp(z_j/\tau)} \le \frac{\exp(z_i/\tau)}{\exp(a/\tau)} = \exp((z_i-a)/\tau) \to_{\tau \to 0} 0$.
> > Therefore, all the coordinates that are smaller than the maximal value will converge to probability $0$ as $\tau$ converges to $0$. It is therefore easy to show when $\tau \to 0$ the vector $s$ converges to a vector that is the uniform distribution over the maximal values of $z$ (which we denote by $\hat{f}_{\text{max}}$).
>
> - To my understanding, in Section 3.4..
>
> > It is possible that our original notation was not clear enough. We denoted by $\delta(\text{condition}) = 1$ the function that outputs $1$ if the condition holds, and wrote $\delta(y \in Y^*_x, x \in X_i)$ to indicate that the function is $1$ if both $y \in Y^*_x$ and $x \in X_i$. Your suggested notation is equivalent but easier to understand, so we will update our paper to use this notation instead.
>
> - In Eq.4..
>
> > Thank you for pointing this out. After correcting, the second expectation in Eq. 4 now correctly reads $y∼f(.∣x)$, which clarifies the calculation of the reward difference.
>
> - In line 237..
>
> > We have made the suggested correction, replacing 'left' with 'right' in line 237 to accurately describe the direction in the context.
> ---
> > We appreciate the feedback and suggestions, which have significantly improved the clarity and rigor of our paper. We hope that these revisions address your concerns and help our paper meet the standards for publication. We believe that our work on transcendence in generative models has the potential to influence future research in this area, and we are grateful for your review and insights. We look forward to further feedback you may have.

---

> > ### Comment · Reviewer_Se7B · 2024-08-14
> > **Thanks for the rebuttal**
> >
> > Thank you for your answers and the additional results and insights provided. After reading also the other reviews and replies, I will raise my score accordingly.

---

### Official Review · Reviewer_mPZQ · 2024-07-12

**Soundness:** 3
**Presentation:** 3
**Contribution:** 3
**Rating:** 6
**Confidence:** 4

**Summary:**

The paper explores the phenomenon where generative models, trained to mimic human behavior, can surpass the performance of the experts generating the training data. The study focuses on autoregressive transformer models trained on chess game transcripts, demonstrating that these models can outperform the best human players in the dataset. The key to this transcendence is identified as low-temperature sampling, which effectively denoises human errors and biases through a majority-voting mechanism. The paper provides theoretical proof and empirical evidence supporting this phenomenon and discusses the necessity of dataset diversity for achieving transcendence.

**Strengths:**

Some Strengths:
1. Novel Concept: The paper introduces and formalizes the concept of transcendence in generative models, providing a new perspective on model performance.
2. Theoretical Foundation: The authors offer rigorous theoretical proofs that support their claims, grounding the concept of transcendence in solid mathematical principles.
3. Empirical Validation: The study includes extensive experiments with chess models, demonstrating the practical applicability of their theoretical findings.
4. Comprehensive Analysis: The paper not only shows that transcendence is possible but also delves into the mechanisms (low-temperature sampling) and conditions (dataset diversity) required to achieve it.

**Weaknesses:**

Here are some weaknesses:
1. Limited Scope: The empirical validation is confined to chess, a well-defined and constrained domain. It remains to be seen if transcendence can be generalized to other, more complex tasks.
2. Assumptions: The theoretical framework relies on several simplifying assumptions, such as uniform sampling of experts and the nature of reward functions, which may not hold in real-world scenarios.
3. Practical Implications: While the concept of transcendence is theoretically and experimentally validated, the practical implications and real-world applications of this phenomenon are not thoroughly explored.
4. Computational Resources: The training of large transformer models, particularly with extensive datasets and multiple temperature settings, requires significant computational resources, potentially limiting reproducibility for researchers with fewer resources.

**Questions:**

1. How do the authors plan to test the concept of transcendence in domains beyond chess, particularly in more complex and less structured tasks?
2. What are the potential impacts of relaxing some of the theoretical assumptions made in this study? For example, how would a non-uniform sampling of experts affect the results?
3. What ethical implications might arise from models that can transcend their training data, particularly in sensitive domains like healthcare or finance?
4. How can dataset diversity be ensured or measured effectively in other domains to facilitate transcendence?
5. Can the authors provide specific examples or case studies where transcendence could lead to significant real-world benefits?

**Limitations:**

1. Domain Specificity: The study is currently limited to chess, and its applicability to other domains is speculative at this point. Further research is needed to confirm the generalizability of the findings.
2. Simplifying Assumptions: The theoretical results are based on several simplifying assumptions that may not hold in practice. Future work should aim to relax these assumptions and test the robustness of the findings.
3. Resource Intensity: The models and experiments require substantial computational resources, which may not be accessible to all researchers, potentially hindering reproducibility and further investigation.
4. Ethical and Practical Implications: The paper briefly touches on the broader impact but does not delve deeply into the ethical and practical implications of deploying transcendent models in real-world applications.
5. Future Work: While the paper lays a strong theoretical foundation, it leaves many avenues for future research, including exploring other domains, addressing ethical concerns, and developing practical applications.

---

> ### Author Rebuttal · Authors · 2024-08-06
>
> > We thank the reviewer for praising the novelty of the concept of transcendence and the rigor of our theoretical proofs while noting concerns about the limited scope of our experiments and the lack of exploration of the practical implications of transcendence.
> ---
> Limited..
>
> > To help address this concern, we have run a new preliminary experiment on the Stanford Question Answering Dataset ([SQuAD 2.0](https://rajpurkar.github.io/SQuAD-explorer/)), testing the effects of temperature denoising performance on several LLMs of various sizes. For more details, please see the main rebuttal PDF and the first rebuttal to Reviewer Se7B.
>
> Assumptions..
>
> >  We acknowledge this both in the theory section itself and in the limitations section at the end. As with many theoretical works in the field, simplifying assumptions is necessary for proving strong results, and these may not always capture the complex nature of real-world settings. To alleviate this problem, we have performed thorough empirical experiments to show that our theoretical findings indeed apply to real-world settings.
>
> Practical..
>
> > While we agree that practical implications and real-world applications are not thoroughly explored, doing so would exceed the scope of our work and lead to a less focused paper. The 9-page limit constrains us from thoroughly exploring theoretical validation, empirical analysis, and practical applications of Transcendence.
>
> > For preliminary discussion, we can identify tasks that benefit most from AI, like copyediting, by considering those with errors aligning with Transcendence's theoretical conditions. Generally, AI excels in situations where the "wisdom of the crowds" surpasses individual judgment, such as question-answer platforms like StackOverflow or detecting spam in Gmail. These domains already leverage AI effectively (e.g., LLMs for copyediting and QA, Naive Bayes for spam).
>
> > [1] "Studying the 'Wisdom of Crowds' at Scale." Proceedings of the AAAI Conference on Human Computation and Crowdsourcing, vol. 7, no. 1, 2019, pp. 171-179.
>
> Computational..
>
> > We would like to kindly clarify a misunderstanding here. We do not use significant computational resources, and training our model requires only a single consumer-level GPU. As detailed in section 4.1, our transformer model is only 50M parameters, which is very small compared to large transformers such as GPT-3 (175B).
>
> How do..
>
> > As noted above, we additionally have new experiments validating that denoising improves performance in Natural Language Question-Answering and Reading Comprehension. Here, it is clear how evaluation is done as human labels are provided. A baseline can easily be measured by having humans perform the task and using their performance as the threshold needed for transcendence. In tasks where human labels cannot be assumed, Elo or Glicko-2 Ratings is a powerful metric for any task where two models can be compared against, which is a much weaker assumption than assuming gold labels or the ground truth is known. In fact, our experiments are run with Glicko-2 Ratings, demonstrating a clear path for measuring transcendence in future more complex tasks.
>
> What are..
>
> > Relaxing some of the theoretical assumptions such as uniform sampling of experts, would in fact potentially enable other forms of transcendence. To give one example, imagine that each expert has a domain of expertise where they are sampled more often and are almost always correct, and outside that domain, they are almost always wrong. Thus, whilst we had $\overline f(y|x) = 1/k \sum_i^k f_i(y|x)$ before, we would now have $\overline f(y|x) = \sum_i^k f_i(y|x) p(i|x)$, where $p(i|x)$ is the prior likelihood. Intuitively, this would enable a new form of transcendence, as the mixture distribution would "choose" the correct expert given different domains, improving performance over any one expert. We leave a more formal exploration of this form of transcendence for future work.
>
> What ethical..
>
> > It is hard to address ethical implications without speculating well beyond the empirical and theoretical results currently available within our paper and related work, and we would not feel comfortable making claims that cannot be backed with rigor or supporting evidence.
>
> How can dataset..
>
> > There have been several works published on measuring and ensuring dataset diversity, using information-theoretic [1] and geometric approaches [2,3].
>
> > [1] Dieng, Adji Bousso, et al. "The Vendi Score: A Diversity Evaluation Metric for Machine Learning." 2 Jul. 2023.
>
> > [2] Han, Jiyeon, et al. "Rarity Score: A New Metric to Evaluate the Uncommonness of Synthesized Images." 26 June 2022.
>
> > [3] Sajjadi, Mehdi S. M. et al. “Assessing Generative Models via Precision and Recall.” 2018.
>
> Can the authors..
>
> > Pre-trained language models have already demonstrated the potential of transcendence in providing significant real-world benefits, including already cited use cases such as copyediting, question-answering, and spam filtration. In addition, Khan Academy has launched Khanmigo [1], an AI-powered assistant that uses advanced language models to offer personalized education, enabling tailored learning experiences for students at scale.
>
> > [1] https://www.khanmigo.ai/
>
> Limitations Section:
>
> > We have addressed all these limitations in the above responses to the weaknesses and questions you have raised.
>
> ---
> > We are grateful to you for your insightful queries and comments. We hope our clarification and comprehensive responses adequately address the concerns raised and encourage a review of the score assigned. We are confident that the transcendence phenomena and subsequent research agenda could have a significant impact on understanding the capabilities and limitations of future generative models and enable future research on this concept. We kindly request a reevaluation of our work, considering its potential contribution to this area, and thank you for your time and consideration.

---

> > ### Comment · Reviewer_mPZQ · 2024-08-11
> > **Good!**
> >
> > Thank you for your response. I have gained a deeper understanding of your work through your replies and interactions with the reviewers. I will consider your comments and adjust my score accordingly.

---

> > > ### Author Response · Authors · 2024-08-13
> > > **Response**
> > >
> > > Thank you for taking the time to engage with our work and for considering our responses. We appreciate your feedback and are glad that our explanations have clarified aspects of the paper. Should you have any further questions or require additional clarifications, we are happy to provide more details. We look forward to your response and await an adjusting of your score.

---

### Author Rebuttal · Authors · 2024-08-06

# Global Note

**Thank You.** We thank the reviewers for their insightful feedback and comments. We are encouraged to find that the reviewers recognized the paper as novel, introducing "a new perspective on model performance" (mPZQ). We are also grateful that the reviewers appreciated the evaluation of our method in chess, "an experimental setting very smart for this study" (Se7b) and "nicely accessible" (np7r), adding credibility to our findings with "clear, convincing results" and "solid empirical validation" (Se7b). We are pleased to hear that the reviewers found the paper "well-structured and well-written" (Se7B,np7r). We also appreciate that the theoretical section was noted as insightful (Se7b) and "relevant in non-trivial settings" (np7r), and offers "rigorous theoretical proofs that support their claims, grounding the concept of transcendence in solid mathematical principles" (mPZQ).


**New Experiments.**  To address the feedback regarding the generalization of our findings beyond chess, we have conducted two new experiments:

1. **NLP Experiment:**
   We extended our analysis to the Natural Language Processing domain by running experiments on the Stanford Question Answering Dataset (SQuAD 2.0). We tested the effects of temperature denoising on the performance of several large language models (LLMs) of varying sizes. The SQuAD task involves reading comprehension and question-answering based on Wikipedia articles, making it an ideal setting to evaluate the impact of denoising on language models. We measured the exact-match, semantic-match, and F1 scores of the model outputs at different temperatures. The results show that temperature denoising leads to improved performance, corroborating the findings of our chess experiments and providing broader validation of the underlying mechanism of temperature denoising in diverse domains.

2. **Toy Theoretical Model:**
   We developed a toy theoretical model to further study when transcendence is possible. This model involves a classification task with Gaussian input data and linearly separable classes. Experts label the data with noisy versions of the ground truth separator. We trained a linear model on a dataset labeled by random experts and observed the test accuracy for different temperature settings. The synthetic experiments demonstrated that transcendence occurs when expert diversity is high and temperature is low, aligning with our theoretical and empirical analysis in the chess domain.

These additional experiments support the generalizability of our findings and provide a more comprehensive understanding of the conditions under which transcendence can occur in different settings. Please find the new results and detailed analysis attached to the 1-page rebuttal PDF.

---

### Decision · Program_Chairs · 2024-09-25

**Decision:**

Accept (poster)

**Comment:**

The paper studies the phenomenon where a generative model, trained on data from human experts, achieves capabilities that surpass the abilities of those experts. The authors provide a theoretical analysis of the conditions under which transcendence can occur. A theoretical framework is provided that allows to analyse the conditions under which transcendence can occur and it is shown that low-temperature sampling is necessary to observe transcendence. The relevance of the findings is demonstrated on chess. In the rebuttal further experimental results on a question-answering set as well as a theoretical toy-model were added, strengthening the paper.

I agree with the two more elaborated reviews that voted for clear acceptance.